# Manufacturing Bacteriophages (Part 2 of 2): Formulation, Analytics and Quality Control Considerations

**DOI:** 10.3390/ph14090895

**Published:** 2021-09-02

**Authors:** Carolina Moraes de Souza, Tayfun Tanir, Marvin Orellana, Aster Escalante, Michael Sandor Koeris

**Affiliations:** Amgen Bioprocessing Center, Department of Biological Engineering and Management, Henry E. Riggs School of Applied Life Sciences, Keck Graduate Institute, Claremont, CA 91711, USA; CMORAESDESO18@students.kgi.edu (C.M.d.S.); ttanir20@students.kgi.edu (T.T.); MORELLANA20@students.kgi.edu (M.O.); Aster_Escalante@kgi.edu (A.E.)

**Keywords:** bacteriophage, phage therapy, manufacturing, formulation, analytical, drug product, fill-finish, drug substance

## Abstract

Within this second piece of the two-part series of phage manufacturing considerations, we are examining the creation of a drug product from a drug substance in the form of formulation, through to fill-finish. Formulation of a drug product, in the case of bacteriophage products, is often considered only after many choices have been made in the development and manufacture of a drug substance, increasing the final product development timeline and difficulty of achieving necessary performance parameters. As with the preceding review in this sequence, we aim to provide the reader with a framework to be able to consider pharmaceutical development choices for the formulation of a bacteriophage-based drug product. The intent is to sensitize and highlight the tradeoffs that are necessary in the development of a finished drug product, and to be able to take the entire spectrum of tradeoffs into account, starting with early-stage R&D efforts. Furthermore, we are arming the reader with an overview of historical and current analytical methods with a special emphasis on most relevant and most widely available methods. Bacteriophages pose some challenges that are related to but also separate from eukaryotic viruses. Last, but not least, we close this two-part series by briefly discussing quality control (QC) aspects of a bacteriophage-based product, taking into consideration the opportunities and challenges that engineered bacteriophages uniquely present and offer.

## 1. Introduction

Historically, much of the formulation process for phages has been an adaptation of what is already known for protein-based therapeutics [1]. Considering the broad possible applications for phages in the clinical setting, it is important to think about how to ensure the best delivery of the phages to their target site. The formulation plays a key role in drug delivery, as well as ensuring drug stability over long periods of time, and it is not different for bacteriophage-based therapies. Furthermore, the formulation process should be carefully assessed and characterized when designing the process, as yield losses are inevitable and changes in demand might necessitate changes in scale for upstream and downstream production [2]. In order to choose the optimal ingredients for a phage therapy formulation, on top of considerations regarding the phage itself, ingredients must always be safe to use in the context of a particular delivery mode, e.g., some ingredients can be safe for oral delivery but not for inhalation delivery [1]. Several authors have been investigating the possible formulations that can be designed for phage therapy and how to improve its long-term stability. Table 1 contains a summary of all the different formulations developed for phage therapy that are discussed in this section.

Additionally, formulations can vary widely depending on the indication: for example topical applications might require careful choice in excipients when being applied to compromised skin structures such as those encountered in the course of treating burns, versus having broader options to be tolerable in the case of inflamed but largely intact skin in the case of mild to moderate acne [12]. Successful formulations share common features, including a combination of proteins or amino acids, as well as certain carbohydrates, which act as cryoprotectants. Certain polyols such as glycerol and sorbitol also provide stability at refrigeration and freezing storage conditions [3,10,13,14].

### 1.1. General Considerations

This section is divided by subheadings. It should provide a concise and precise description of the experimental results, their interpretation, as well as the experimental conclusions that can be drawn.

The formulation process is one of the main steps in the drug product manufacturing process. Often, formulation varies according to the desired final dosage form, which can fall in three different general categories: solid, semi-solid and liquid. Distinct formulations might require unique processes, nonetheless there are steps that are present in most cases (Figure 1) [2]. Still in formulation, the process can include specialized treatment such as encapsulation with liposomes or polymers, which are represented in light blue at Figure 1 as options in the formulation process.

Phages are proteinaceous entities and their nature make them inherently sensitive to environmental conditions. A number of factors are known to impact phage structure and infectivity [2,15,16,17,18], and similar to our review focusing on cell-line development, upstream and downstream processes. Among these factors, the most critical to consider for the formulation of a drug product are temperature, pH, ionic strength, adsorption to matrices and shear stress [2,9,18,19,20]. Furthermore, another factor to be carefully considered is their susceptibility to high frequencies. Leung et al. observed such sensitivity when the process of spray-freeze drying was evaluated: bacteriophages are exposed to an ultrasonic nozzle employed for droplet generation, which resulted in a significant titer loss of more than 2 log [8], despite stabilizing excipients and cold temperatures. Additionally, it is important to note that phage morphology varies widely between different phages, and therefore might have distinct physicochemical properties. When developing a formulation for a phage therapy it is important to take into consideration the physicochemical properties of the phage or phages in question. Engineered phages present an opportunity and a problem for the formulation process developer principally in this step as it relates to the physiochemical modification of the phage itself. Within the spectrum of engineering approaches it is common to change the surface charge of the phage, which influences the mixing, aggregation and other behavior with excipients [21]. Further engineering might focus on the structure and electrical surface charge of the tail fibers [22], which deserves special attention. Tail fibers are at risk to agglomerate, be sheared or otherwise damaged in the process of formulation and storage, and while no predictive algorithm exists, it is critical that robust analytics are leveraged to characterize the impact that formulation have, especially on infectivity. For a selection of compositions, refer to Table 2.

### 1.2. Liquid Formulation

Liquid formulations are the simplest available formulas for the delivery of phage therapies. Quite often, it does not require intense investment in formulation development as phages tend to be quite stable in liquid solutions that offer pH control and free ions such as Mg^2+^ and Ca^2+^ [15]. The main issue with liquid formulations is that some of the compounds used are not safe for use in all formulations, such as formulas targeting the respiratory tract [1]. Furthermore, simple formulas might not be useful in oral delivery, as they need to be designed to resist the harsh stomachal acids. One of the possible solutions to this issue is the encapsulation of phages in liposomes [4] or polymeric materials such as alginate. On top of encapsulation, antiacids such as CaCl_2_ can also be added to preserve phage titers in low pH [3]. Liquid formulations for phages currently offer relatively good stability when stored at low temperatures in a refrigerator. A few of the commercial products that are available in Russia and Georgia offer a 1–2-year shelf-life in a liquid formulation stored at a temperature of 2 °C to 8 °C (Eliava Biopreparations, Microgen products). Unfortunately, there are not enough stability data available for these products [1]. Nonetheless, a number of authors have investigated the stability of phages in liquid formulations and their data support stability up to 2 or more years [2].

### 1.3. Encapsulation

Here, the study moves from the general liquid formulation to specific encapsulation formulations. As previously discussed, one of the key elements that must be carefully considered when developing the formulation for a phage therapy aside from drug product stability requirements is the target site of release. One of the most challenging sites for delivery of viable phages is the gastrointestinal (GI) tract, due to its varying pH and the presence of numerous enzymes [3]. One of the solutions that was explored is the encapsulation of phages for enteric release. The purpose of encapsulating the phages is to protect them from the harsh conditions present in reaching the GI tract, as well as enhance their residence time in the gut, including but not limited to adherence to the mucosal lining, which improves the contact between phage and host [3,4]. Numerous techniques have been explored in the last 10 years, being the most prominent the alginate encapsulation and liposome encapsulation [1]. Bacteriophages are mostly negatively charged in order to repel the head group from the negatively charged surface of the bacterial target [23]. Further advantages are easy adherence to mammalian cell surfaces as well as potential bacterial cell surfaces, as well as providing a first electrostatic barrier to proteins from acidic environments such as the stomach of a patient, if enteric release is desired. Lipid nanoparticles degrade rapidly in the presence of bile salts, while exhibiting long-term stability in aqueous conditions under freezing temperatures [24,25]. This feature makes the lipid nanoparticle formulation a good choice for oral delivery [25,26]. However, the physicochemical characteristics of the phage might affect the release kinetics from the liposome. Common buffering parameters are to use magnesium sulfate at pH 6.1 and add trehalose to the liposome solution for improved lyophilization as with other lyophilization approaches [27]. The described encapsulation process has a yield of around 48%, with increased titer post lyophilization for liposome encapsulated phages (22% (13.4) vs 82.3% (15.4); 2% (1.5) vs 15.1% (9.6); 47.5% (12.1) vs 84.4% (16.9)). Colom et al., were able to encapsulate three different phages into cationic liposomes [4]. The liposomes consisted of a mix of four lipids, their composition was such that they held a positive charge, which was highly beneficial as these would easily encapsulate compounds that hold a negative charge, such as phages. A detailed composition of the liposome-encapsulated formulation is present in Table 2. On top of being a good barrier to protons, cationic liposomes tend to adhere to the mucosal lining, as cells also have a negative charge, which aids increasing residence time in the GI tract [4]. Phages that were encapsulated suffered a smaller log-reduction in titer when exposed to a low pH environment than a non-encapsulated phage (encapsulated phages had their titers reduced by 3.7 log to 5.4 log, while non-encapsulated phage titers were reduced by 5.7 log to 7.8 log). Although promising, liposomal encapsulation has its downsides: the efficiency of the encapsulation process is relatively low, holding a yield of about 48% [4]. Another concern is the variability in liposome size which might cause concern with regulatory agencies [28]. Additionally, although liposome encapsulation seems to protect phages during lyophilization, the titers after lyophilization still seem to be strongly phage-species dependent [4,28,29,30]. As with all things pertaining phages, it is paramount to characterize the drug substance thoroughly before making any decisions regarding formulation.

### 1.4. Semi-Solid Formulations

The robust and predictable development of semi-solid formulations for phage therapies, such as creams, gels, and aerosols, expands the potential use of phage therapy. Topical phage therapy can aid the treatment of a number of skin infections which includes acne, infected wounds, pulmonary infections and chronic otitis [31]. According to the US Pharmacopeia (USP), a cream, gel, ointment, and paste are different forms of semi-solid formulations. The difference between them refers to a few attributes. Creams are formulas that have an opaque, viscous, relatively soft, consistently spreadable emulsion that often are more than 20% water and volatiles, and normally less than 50% hydrocarbons, waxes, or polyols as the vehicle for the drug substance. Ointments, on the other hand, are less than 20% water and volatiles, and more than 50% hydrocarbons, waxes, or polyols. Pastes are thick and stiff, with a high concentration of insoluble powder substances (20% to 50%) that are finely dispersed in a fatty or aqueous base. Gels, according to the USP, contain dispersions of small or large molecules in an aqueous-based vehicle rendered jelly-like through the adding of a gelling agent. When developing a semi-solid formula, one important consideration to be made is the composition of the dosage form. There are several bases that can be used for these formulations, but not all are a good fit for phage therapy formulations. For example, when designing a cream formula, it is important to consider which is the target tissue, as well as levels of permeation, no irritation, compatibility with the container-closure system and active pharmaceutical ingredient (API) stability. Additionally, semi-solid formulations are often specific, and it is paramount to understand what are the desired properties, as well as the ones that should be avoided in order to achieve the desired outcomes [32]. Brown observed that for formulating a phage into a cream, it is key to use a base that is non-ionic, as it minimizes the interaction between the base and the phages [31]. Therefore, the charge of the excipients should be taken into consideration to avoid phage inactivation. Another important aspect is the mixing process. Even distribution of the active pharmaceutical ingredient throughout the formula is necessary. A common technique used to achieve a more even distribution is known as geometric dilution, and consists of adding the formula gradually to the drug substance [10]. Additionally, as phages are highly sensitive to shear stress, the mixing process must take that into account. Furthermore, the viscosity of the formula can also be a concern as highly viscous fluids tend to offer higher shear stress, which can cause phage inactivation [2,8,9,18,33,34].

### 1.5. Solid Formulations

Solid formulations are the preferred formulation from the patient standpoint as patients prefer to take pills instead of injections or to spend a few hours with an IV. Solid formulations, nonetheless, are not limited to pills and capsules. Solid formulations also include devices, as these can be coated with a drug. Bandages and catheters that have the drug adsorbed in their structure are another promising formulation for phage therapy. Curtin et al., described the adsorption of a bacteriophage against *Staphylococcus epidermidis* in catheters using a hydrogel-coated silicone catheter, which prevented biofilm formation [7]. Immobilizing phages in surfaces requires specific techniques that preserve phage activity. Hosseindoust et al., evaluated three distinct binding methods: physisorption, polyelectrolyte adsorption using poly-L-lysine, and covalent binding using glutaraldehyde cross-linking [6]. When comparing the three methods they observed that covalent binding minimized detachment. Additionally, the group observed that the orientation in which the phage was adsorbed in the surface interfered with phage infectivity, and is something to take into consideration when immobilizing phages for phage therapy [6].

For solid formulations such as pills and capsules, it is necessary to dry the phage solution. There are a number of drying techniques that are used to dry phage formulations, which include lyophilization, spray-drying and spray-freeze drying [2]. Additionally, a granulation process is also required to mix all powder excipients that are added to compose the tablets. There are multiple modalities available, but for drug substances that often struggle with stability issues it is recommended the use of dry granulation [35]. Alternative solid formulations such as suppositories and troches were also investigated for phage therapy [10]. The process did not require drying, and the formula was produced using a liquid phage concentrate. The suppository and troche formulation were shown to be stable for 49 days and 56 days, respectively, when stored at a temperature of 4 °C. Lytic activity was still present, but a 0.6 log reduction in titer was observed [10]. The detailed composition of both formulas are available in Table 2.

### 1.6. Scale-Up Considerations

Similar to drug substance (DS), drug product (DP) manufacturing is also subject to scale-up challenges. When the API has a proteinaceous nature as in the case of bacteriophages, process parameters must be carefully controlled during formulation, fill, and finish to preserve the product and maximize yield. Quite often most challenges occur during the drying processes, in particular when there are freeze-drying steps [36]. Drying processes involve complex heat and mass transfer phenomena, which are difficult to model and can make scale-up unpredictable; this is further compounded by a lack of predicate bacteriophage processes that were run at scale. For most drying processes it is important to remember that different load sizes, chamber pressure, ramp rate and shelf temperature are parameters known to impact the performance of the process [37,38]. Additionally, lyophilizers can be a cost-intensive unit operation from both the investment and operational side, further making optimization and proper sizing key to ensure a cost-effective process [39]. Historically, freeze and spray-drying processes have been optimized using a trial-and-error strategy, which is not only time but resource consuming. A number of authors emphasize the necessity to take a systematized approach to drying processes but no generalized framework exists [37,39,40].

Spray-drying in contrast to freeze-drying has been much less widespread in the biological API preparation domain, principally due to concerns about thermal degradation during the drying process. However, compared with traditional freeze-drying processes, spray-drying offers several advantages, particularly when it comes to scalability. Starting with the mode of operation, spray-drying processes can be operated in a continuous mode, while freeze-drying is necessarily a batch operation. The advent of high pressure spray-dryers mitigate thermal exposure via a very high flow rate [40]. Finally, compared with lyophilizers, spray driers offer more options for control and metrology, as it is possible to modify the cycle mode, atomizer type and airflow patterns to fit the needs of the final product [41]. In several studies, spray-drying had a superior performance to freeze-drying [8,9,34]. A word of caution for the newly converted spray-drying enthusiast, however: for the particular case of phages, it also is important to pay close attention to the impact of the high frequencies present in the process, considering that those factors can have a detrimental effect over yield, as observed by Leung et al. [8].

## 2. Analytical

Analytical testing is one of the key elements to invest in to develop a successful phage manufacturing process. At every step of the process, all the way from cell-line development to formulation, fill and finish, it is critical to be able to verify that the drug substance and drug product are meeting specifications. Additionally, the analytical assays used throughout the manufacturing process, all the way to release testing, must be well-established and validated. Therefore, it is critical to ensure assay repeatability, sensitivity, and accuracy (International Council for Harmonization of Technical Requirements for Pharmaceuticals for Human Use (ICH) Q6B; US Food and Drug Administration (FDA) Guidance for Industry Analytical Procedures and Methods Validation for Drugs and Biologics [42]). Recommendations made by the FDA on testing biotechnology products can be found in their guidance for analytical procedures and method validation for biologics.

Testing products that are composed by phages can be challenging as most of the historical and well-established methods for infectivity and quantification are laborious and time consuming [43,44,45]; nonetheless, in the last ten years, several new methods that take advantage of molecular biology techniques have gained increasingly widespread use. Table 3 contains a summary of the methods available for testing phages covering the four key areas that must be evaluated when developing a therapeutic product.

Several groups have been investigating quantitative polymerase chain reaction (qPCR) as a method for phage quantification. Duyvejonck et al. developed a method for the quantification of phages in a phage cocktail utilizing qPCR [43]. The group compared their method to the traditional agar overlay method, which is the standard for phage quantification. Using qPCR might offer a much faster, cost-effective analytical platform for phage titration. In addition to being faster, this method offers high reproducibility. Compared with the traditional method, qPCR presented a smaller coefficient of variation, as well as smaller inter- and intra-operator variability [43]. In 2012, Refardt et al., also investigated qPCR as a tool for quantification, as well as identification of phages. They observed that this method has high reliability as it provided low variation between replicates [46]. Consequently, qPCR can be a great tool for phage quantification and identification. It can potentially serve multiple attributes for in process and release testing, as it is a fast, specific, and reliable method.

## 3. Quality Control

The quality control (QC) of bacteriophages, such as that of all biopharmaceuticals is aimed to demonstrate their safety, efficacy, identity, and purity utilizing analytical methods that are validated based upon the regulatory guidelines. In the case of phage quality control, at any level of production where exposure to the environment takes place, the quality and cleanliness of the air must be grade A. Since only the host bacteria have the ability to infect humans, the pathogenicity of the host bacteria determines the biosafety level (BSL) [51]. Both master banks must be checked for identity, purity (both banks must include only the desired host bacteria, and phage(s)), viability for master cell bank (MCB) and biological function for master phage bank (MPB). The MPB should be made sure of a lack of virulent genes and prophages by exploiting state-of-the-art DNA/RNA sequencing and bioinformatics techniques, meaning that next-generation sequencing (NGS) should be used to confirm the identity of the phage(s) in the MPB [51]. Lytic phages are recommended to be utilized in phage therapy production because lysogenic phages can potentially transfer virulent genes in the host bacteria to other bacterial strains [52]. The bacteria to be used in the production must be checked for identity using molecular typing techniques such as multi-locus sequence typing (MLST) [53,54], repetitive polymerase chain reaction (PCR), amplified fragment length polymorphism (AFLP), and pulsed-field gel electrophoresis (PFGE) [51]. The spontaneous mutations remain inevitable during the production phase, the number of mutations should be limited to a level which they do not alter the biology and function of the phage. The acceptable amount of sequence difference between the MPB and phage in the production stage should be determined in a case-by-case manner [55]. For the QC of the phage DP, identity, general purity, the titters, bacterial contaminants such as endotoxin, nucleic acid contaminants, and bacterial proteins should be tested with the appropriate tests. For identity, as for master seed lots, the NGS technique is recommended [51]. As mentioned previously, a DP of phage therapy can be comprising of either a single phage or multiple phages (phage cocktail). To determine the titter of each of the phages, time-kill assay can be used as well as the double-agar layer (DAL) technique [56]. Whilst qPCR and enzyme-linked immunosorbent assay (ELISA) can also be used to determine titters, they check for single components unlike DAL and time-kill assay. For the quality control of engineered phages, it is required to perform an environmental risk analysis to evaluate the risk of transmission of the genetically modified organism (GMO) to other organisms [57,58].

## 4. Summary

In conclusion, it is paramount to consider the journey of bringing a bacteriophage-based product to market in its entirety and feed information from parallel experimentation into the development process. The authors are of the opinion that process development begins before a finalized research-grade product exists, such that cell-line development, upstream, downstream, and formulation considerations are considered and inform the final drug product.

## Figures and Tables

**Figure 1 pharmaceuticals-14-00895-f001:**
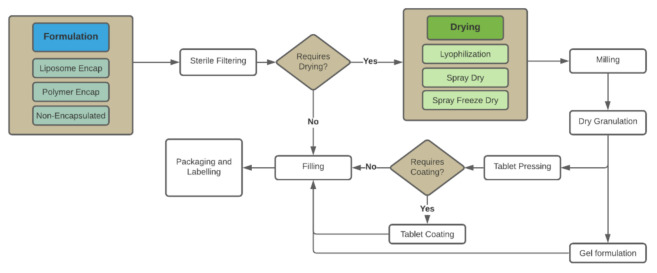
Process flow diagram for formulation, fill and finish. The drug substance (DS) is bulk formulated and sterile filtered. From there, the process developer chooses the necessary drying steps or fills in liquid, or semi-liquid (gel) format.

**Table 1 pharmaceuticals-14-00895-t001:** Illustration of formulations reported in the literature for phage therapy.

Formulation	References
**Solid**	
Pill	Gonzalez-Menendez 2018 [3]; Colom 2015 [4]
Capsule	Richards 2021 [5]
Bandage	Hossendoust 2011 [6]
Device	Curtin 2006 [7]
Aerosol	Leung 2016 [8]; Matinkhoo 2011 [9]
Troche	Brown 2017 [10]
Suppository	Brown 2017 [10]
**Semi-Solid**	
Cream	Brown 2017 [10]
Gel	Alfadhel 2011 [11]
Ointment	Brown 2017 [10]
Paste	Brown 2017 [10]
**Liquid**	
Injection	Malik 2017 [2]
Infusion	Malik 2017 [2]
Aerosol	Leung 2016 [8]

**Table 2 pharmaceuticals-14-00895-t002:** Summary of the methods and formulations used for different phages. Gonzalez-Menendez et al., compared distinct formulas between four different phages, two from the Myoviridiae family and two from the Siphoviridiae family [3]. Their stability over a period of 24 months varied across the formulas but were most markedly between the two families. Phage cocktails offer additional challenges as it is necessary to find a formula that works well for all phages present in the cocktail [9].

Phage	Formulation	Method	Composition	Reference
PEV2	Solid	Spray-Freeze Dry or Spray-Dry	Salt-magnesium buffer (SMB, 5.2 g/L sodium chloride, 2 g/L magnesium sulfate, 6.35 g/L Tris-HCL, 1.18 g/L Tris base, 0.01% gelatin solution, with pH adjusted to 7.5), D-(+)-Trehalose dihydrate (60% and 40% *w*/*v*), mannitol (20% and 40% *w*/*v*) and L-leucine (20% *w*/*v*).	Leung 2016 [8]
UAB_Phi20, UAB_Phi78, or UAB_Ph87	Solid-Liposome Encapsulated	Lyophilized	3.2% *w*/*v* trehalose, MgSO_4_ (10 mM, pH 6.1), lipid mixture of 1,2-dilauroyl-racglycero- 3-phosphocholine (DLPC), cholesteryl polyethylene glycol 600 sebacate (Chol-PEG600), cholesterol (Chol), and cholesteryl 3-*N*-(dimethylaminoethyl) carbamate hydrochloride (cholesteryl) (1:0.1:0.2:0.7 molar ratio).	Colom 2015 [4]
KOX1	Solid–Suppository	Mixed	Phage concentrated in PBS. Suppository composed by gelatin powder, purified water and Glycerol. Final concentration at 4.5 × 10^8^ PFU/g.	Brown 2017 [10]
KOX1	Solid–Troche	Mixed	Phage concentrated in PBS. Silica gel (micronized), stevioside powder, acacia powder and citric acid anhydrous powder, polyethylene glycol base. Phage final concentration 4.5 × 10^8^ PFU/g.	Brown 2017 [10]
phiIPLA35, phiIPLA88, phiIPLA-RODI and phiIPLA-C1C	Liquid	Mixed	SM buffer (20 mM Tris HCl, 10 mM MgSO_4_, 10 mM Ca(NO_3_)_2_ and 0.1 M NaCl, pH 7.5), 0.8 M trehalose, 0.8 M sucrose, 15% glycerol or 11% skim milk, final titer ranging from 10^8^ to 10^9^ PFU/mL.	Gonzalez-Menendez 2018 [3]
phiIPLA35, phiIPLA88, phiIPLA-RODI and phiIPLA-C1C	Solid	Lyophilized	SM buffer (20 mM Tris HCl, 10 mM MgSO_4_, 10 mM Ca(NO_3_)_2_ and 0.1 M NaCl, pH 7.5), 22% skim milk, 1.6 M sucrose or 30% sorbitol.	Gonzalez-Menendez 2018 [3]
phiIPLA35, phiIPLA88, phiIPLA-RODI and phiIPLA-C1C	Solid-Alginate Encapsulated	Droplet Encapsulation	SM buffer (20 mM Tris HCl, 10 mM MgSO_4_, 10 mM Ca(NO_3_)_2_ and 0.1 M NaCl, pH 7.5), 50 mM HEPES pH 7.5 containing 2% (*w*/*v*) sodium alginate, 0.1 M CaCl_2._	Gonzalez-Menendez 2018 [3]
phiIPLA35, phiIPLA88, phiIPLA-RODI and phiIPLA-C1C	Solid-Alginate Microencapsulated	Emulsification	SM buffer (20 mM Tris HCl, 10 mM MgSO_4_, 10 mM Ca(NO_3_)_2_ and 0.1 M NaCl, pH 7.5), 50 mM HEPES pH 7.5 containing 3% (*w*/*v*) sodium alginate, 30 mM CaCl_2_, Miglyol 812 containing 3% (*w*/*v*) Span 80, and 50 μL of glacial acetic acid.	Gonzalez-Menendez 2018 [3]
phiIPLA35, phiIPLA88, phiIPLA-RODI and phiIPLA-C1C	Solid	Spray Dried	SM buffer (20 mM Tris HCl, 10 mM MgSO_4_, 10 mM Ca(NO_3_)_2_ and 0.1 M NaCl, pH 7.5), trehalose (15% final concentration) or skim milk (11% final concentration).	Gonzalez-Menendez 2018 [3]

**Table 3 pharmaceuticals-14-00895-t003:** Methods that can be used to evaluate product attributes for purposes of release testing as well as in process control testing.

Area	Method	Attribute	Potential Use	Reference
Potency	qPCR	Adventitious phages	Release Testing; In Process	Refardt, 2012 [46]
		Quantification of phage particles	Release Testing; In Process	Refardt, 2012 [46]; Immamovic, 2010 [47]; uyvejonck, 2019 [43]
		Infectivity	Release Testing; In Process	Immamovic 2010 [47]
	Imaging Flow Cytometry	Determination of phage viability	Release Testing; In Process	Yang, 2019 [48]
		Quantification of phage particles	Release Testing; In Process	Yang, 2019 [48]
	Double-Agar Overlay	Infectivity	Release Testing	Sanders, 1991 [44]
Identity	qPCR	Phage identification	Release Testing; In process	Refardt, 2012 [46]; Immamovic, 2010 [47]; Duyvejonck, 2019 [43]
	Fluorescence-activated Single Cell Sorting (FACS)	Phage identification	Release Testing; In process	Yang, 2019 [48]
	Multiplex PCR	Phage identification	Release Testing; In process	Delrio [49]
Purity	qPCR	Identification of adventitious agents	Release Testing; In process	Duyvejonck, 2019 [43]
Safety	qPCR	Presence of latent phages	Release Testing; In process	Refardt, 2012 [46]
	End-point chromogenic Limulus Amebocyte Lysate	Amount of endotoxins present	Release Testing; In process	Szermer-Olearnik 2015 [50]

## Data Availability

Data sharing not applicable.

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
