# Peer review of "Manufacturing Bacteriophages (Part 2 of 2): Formulation, Analytics and Quality Control Considerations"

_pharmaceuticals, 2021, doi:10.3390/ph14090895_

Round 1

Reviewer 1 Report

Phage therapy has raised in the last years as a good alternative to fight AMR, however the widespread use of phages as viable alternatives to antibiotics in human therapy is still restricted due to the lack of regulatory guidelines and the lack of unknown data of phage application in humans. Therefore, it is important to explore and demonstrate the safety of phages in clinical applications.

This review tackles an important feature for the use of phages for phage therapy, however, in addition to the different phage formulations, I miss in this review more details about the ease at which their production can be scaled up and about the fact that different formulation are required for different applications.

After the authors write something about this and address the minor issues outlined below, I believe it can be published.

Minor issues:

  • Line 3: in the title write “quality control” instead of QC.
  • Line 25: separate keywords by “;” or “,” not both.
  • Line 37: just write “optimal” not “more optimal”.
  • Line 41: “could be designed”.
  • Line 49: “options to be tolerable”.
  • Line 57: you star this section with the sentence “This section may be divided” but I do not understand why. You should state that “This section is divided”, I am right?
  • Line 68-70: remove this 3 lines, it is the figure 1 caption so you do not need to state it in the text. The same in lines 74-76.
  • Line 71: there is a “.” before the beginning of the sentence.
  • Line 77: this is a lost sentence.
  • Line 84: “they also appear”, remove “are”.
  • Line 85: “such sensitivity”, remove “a”. Write et al in italic.
  • Line 87: add a space between log and the reference.
  • Line 96: add a space between excipients and the reference.
  • Line 97: add a space between fibers and the reference.
  • Line 99: “it is critical that the robust analytics are”.
  • Line 101: In the figure caption write as you wrote in figure 1: “Summary of the methods and formulations used for different phages”.
  • Line 102: et al in italic.
  • Line 104: “but were most markedly between”.
  • Line 122: the degree Celsius is oC not ºC, please change. Remove the “.” before the brackets.
  • Line 133: write in full what GI means and then put (GI). Add a space between enzymes and the reference.
  • Line 140: add a space before the reference.
  • Line 153: et al in italic.
  • Line 158: add a space before the reference.
  • Line 178: “spreadable emulsion”, remove the “,”.
  • Line 183: state what USP means.
  • Line 190: state what API means.
  • Line 199: add a space before the reference.
  • Line 212: Staphylococcus epidermidis. Write the full name in the first mention, in italic, then you can just write epidermidis. Write et al in italic.
  • Line 213: remove “.” before the reference.
  • Line 215: et al in italic.
  • Line 217: add a space before the reference.
  • Line 228: add a space before the reference.
  • Line 230: write the degree Celsius as o
  • Line 232: check this error.
  • Line 240: write what ICH Q6B and FDA means.
  • Line 247: after this line there is a table without caption, or mention in the text!
  • Line 248: the beginning of the sentence is missing.
  • Line 249-250: et al in italic.
  • Line 256: et al in italic.
  • Line 261-262: this is the table mentioned above that is between lines 247 and 248. It should be table 3 not 1. Please correct this.
  • Line 271: add a space before the reference.
  • Lines 272-273: the abbreviation MBCB do not correspond for what you wrote: Master Cell Bacteria Bank and the MPBC meaning is missing.
  • Line 275: write what NGS means then put (NGS).
  • Line 278: add a space before the reference.
  • Line 280: the first mention of anything you should write it in full than use abbreviations. Write what MLST, AFLP and PFGE means.

In table 3 (which you erroneous write as Table 1), write what the abbreviations means, in the first mention of qPCR and FACS. Tables 2 and 3 are not mentioned in the text. When you write et al sometimes you put a dot “.”at the end and other times you don’t, be consistent.

Author Response

Phage therapy has raised in the last years as a good alternative to fight AMR, however the widespread use of phages as viable alternatives to antibiotics in human therapy is still restricted due to the lack of regulatory guidelines and the lack of unknown data of phage application in humans. Therefore, it is important to explore and demonstrate the safety of phages in clinical applications.

This review tackles an important feature for the use of phages for phage therapy, however, in addition to the different phage formulations, I miss in this review more details about the ease at which their production can be scaled up and about the fact that different formulation are required for different applications.

Addressed in the manuscript (see attached)

After the authors write something about this and address the minor issues outlined below, I believe it can be published.

Minor issues:

  • Line 3: in the title write “quality control” instead of QC.
    • Resolved, line 3
  • Line 25: separate keywords by “;” or “,” not both.
    • Resolved, line 25
  • Line 37: just write “optimal” not “more optimal”.
    • Resolved, line 37
  • Line 41: “could be designed”.
    • Resolved, line 41
  • Line 49: “options to be tolerable”.
    • Resolved, line 49
  • Line 57: you star this section with the sentence “This section may be divided” but I do not understand why. You should state that “This section is divided”, I am right?
    •  Resolved, line 57
  • Line 68-70: remove these 3 lines, it is the figure 1 caption, so you do not need to state it in the text. The same in lines 74-76.
    • Resolved, line 66
  • Line 71: there is a “.” before the beginning of the sentence.
    • Resolved, line 69
  • Line 77: this is a lost sentence. - The phrase was adjusted to improve the flow.
    • Resolved, line 77
  • Line 84: “they also appear”, remove “are”.
    • Resolved, line 78
  • Line 85: “such sensitivity”, remove “a”. Write et al in italic.
    • Resolved, line 78
  • Line 87: add a space between log and the reference.
    • Resolved, line 81
  • Line 96: add a space between excipients and the reference.
    • Resolved, line 89
  • Line 97: add a space between fibers and the reference.
    • Resolved, line 90
  • Line 99: “it is critical that the robust analytics are”.
    • Resolved, line 93
  • Line 101: In the figure caption write as you wrote in figure 1: “Summary of the methods and formulations used for different phages”.
    • Resolved, line 95
  • Line 102: et al in italic.
    • Resolved, line 96
  • Line 104: “but were most markedly between”. - Resolved, line 98
  • Line 122: the degree Celsius is oC not ºC, please change. Remove the “.” before the brackets.
    • Resolved, line 116
  • Line 133: write in full what GI means and then put (GI). Add a space between enzymes and the reference.
    • Resolved, line 127
  • Line 140: add a space before the reference.
    • Resolved, line 134
  • Line 153: et al in italic.
    • Resolved, line 147
  • Line 158: add a space before the reference.
    • Resolved, line 154
  • Line 178: “spreadable emulsion”, remove the “,”.
    • Resolved, line 174
  • Line 183: state what USP means.
    • Resolved, line 171
  • Line 190: state what API means.
    • Resolved, line 186
  • Line 199: add a space before the reference.
    • Resolved, line 196
  • Line 212: Staphylococcus epidermidis. Write the full name in the first mention, in italic, then you can just write epidermidis. Write et al in italic.
    • Resolved, line 208
  • Line 213: remove “.” before the reference.
    • Resolved, line 210
  • Line 215: et al in italic.
    • Resolved, line 211
  • Line 217: add a space before the reference.
    • Resolved, line 213
  • Line 228: add a space before the reference.
    • Resolved, line 224
  • Line 230: write the degree Celsius as o
    • Resolved, line 226
  • Line 232: check this error.
    • Resolved, line 228
  • Line 240: write what ICH Q6B and FDA means.
    • Resolved, line 269, 270
  • Line 247: after this line there is a table without caption, or mention in the text!
    • Resolved, line 277
  • Line 248: the beginning of the sentence is missing. - Broken into paragraph to improve the text flow.
    • Resolved, line 280
  • Line 249-250: et al in italic.
    • Resolved, line 281
  • Line 256: et al in italic. - Resolved, line 287
  • Line 261-262: this is the table mentioned above that is between lines 247 and 248. It should be table 3 not 1. Please correct this.
    • Resolved, line 293-294
  • Line 271: add a space before the reference.
    • Resolved, line 303
  • Lines 272-273: the abbreviation MBCB do not correspond for what you wrote: Master Cell Bacteria Bank and the MPBC meaning is missing. - The abbreviations were not written correctly. 
    • Resolved, line 304-305
  • Line 275: write what NGS means then put (NGS).
    • Resolved, line 307
  • Line 278: add a space before the reference.
    • Resolved, line 310
  • Line 280: the first mention of anything you should write it in full than use abbreviations. Write what MLST, AFLP and PFGE means.
    • Resolved, line 312-314
  • In table 3 (which you erroneous write as Table 1), write what the abbreviations means, in the first mention of qPCR and FACS. Tables 2 and 3 are not mentioned in the text.
    • Both tables were mentioned in the text, but table 2 had one mention further in the text. To improve the connection a few more mentions were added in parts of the text where it makes sense to refer to the table.
    • Resolved, lines 228, Table 3.
  • When you write et al. sometimes you put a dot “.”at the end and other times you don’t, be consistent.
    • Resolved throughout.

Reviewer 2 Report

 In the manuscript “Manufacturing bacteriophages (Part 2 of 2): formulation, analytics, and QC considerations” of Carolina Moraes de Souza and colleagues, the author had carefully and comprehensively reviewed the creation of drug product from drug substance in the form of formulation through to fill-finish. The authors provided a broad overview of historical and current analytical methods, which are many used in the community. Furthermore, in the review quality control aspects of a phage-based product were discussed.

It is the opinion of this reviewer that the review presented here is outstanding. The paper is well written and organized and comprises all necessary information that are relevant for the topic. The manuscript can be accepted for publication only with minor spelling and style error corrections.

Line 71: remove the dot at the beginning of the sentence

Line 122: please remove the dot before the reference

Author Response

In the manuscript “Manufacturing bacteriophages (Part 2 of 2): formulation, analytics, and QC considerations” of Carolina Moraes de Souza and colleagues, the author had carefully and comprehensively reviewed the creation of drug product from drug substance in the form of formulation through to fill-finish. The authors provided a broad overview of historical and current analytical methods, which are many used in the community. Furthermore, in the review quality control aspects of a phage-based product were discussed.

It is the opinion of this reviewer that the review presented here is outstanding. The paper is well written and organized and comprises all necessary information that are relevant for the topic. The manuscript can be accepted for publication only with minor spelling and style error corrections.

  • Line 71: remove the dot at the beginning of the sentence.
    • Resolved, line 69
  • Line 122: please remove the dot before the reference.
    • Resolved, line 116